Enhancing drought resilience in durum wheat: effect of root architecture and genotypic performance in semi-arid rainfed regions

Boudiar Ridha r.boudiar@crbt.dz boudiarreda@yahoo.fr 1
Mekhlouf Abdelhamid 2
Bekkar Yacine 2
Yessaadi Meriem 2
Bachir Adel 3 4
Karkour Larbi 1
Casas Ana Maria 5
Igartua Ernesto igartua@eead.csic.es 5
1 Department of Biotechnology & Agriculture, Biotechnology Research Center-C.R.Bt-Constantine , El Khroub , Algeria
2 Laboratoire d’Amélioration et de Développement de la Production Végétale et Animale (LADPVA), University of Ferhat ABBAS (UFAS-Sétif1) , Sétif , Algeria
3 Agricultural Experimental Station, Field Crop Institute (ITGC) , Sétif , Algeria
4 Laboratoire de Phytopathologie et Biologie Moléculaire, Ecole Nationale Supérieure d’Agronomie , Algiers , Algeria
5 Estación Experimental de Aula Dei, EEAD-CSIC , Zaragoza , Spain
Abd El-Moneim Diaa
Electronic publication date: 2025 Mar 27
Publication date: 2025
Volume: 13
Electronic Location ID: e19096
Received 2024 Oct 15; Accepted 2025 Feb 11
Copyright: ©2025 Boudiar et al.
Copyright year: 2025
Copyright holder: Boudiar et al.
License: This is an open access article distributed under the terms of the Creative Commons Attribution License, which permits unrestricted use, distribution, reproduction and adaptation in any medium and for any purpose provided that it is properly attributed. For attribution, the original author(s), title, publication source (PeerJ) and either DOI or URL of the article must be cited.
License URL: https://creativecommons.org/licenses/by/4.0/

Keywords: Drought, Root traits, Landraces, Modern cultivars, Root system architecture

Funding: Programme National Exceptionnel PNE 2018/2019 (Ministere d’Enseignement Superieur et de la Recherche Scientifique, Algeria) Agencia Estatal de Investigación PID2019-111621RB-100 This research is part of Ridha Boudiar’s PhD thesis, whose stay at EEAD was supported by a grant from the Programme National Exceptionnel PNE 2018/2019 (Ministere d’Enseignement Superieur et de la Recherche Scientifique, Algeria), and by project PID2019-111621RB-100, granted by the Agencia Estatal de Investigación from Spain. The funders had no role in study design, data collection and analysis, decision to publish, or preparation of the manuscript.

==============================
Background

Developing drought-adapted genotypes is a primary goal for achieving resilient agriculture in the Mediterranean region. Durum wheat, a widely grown crop in the drylands of the Mediterranean basin, would significantly benefit from increased drought resistance.

Methods

We investigated a diverse set of 30 durum wheat varieties, including both local landraces and modern cultivars that have proven successful in Algeria. These varieties were evaluated in field trials over two consecutive years with contrasting rainfall patterns (one very dry, the other quite wet). Grain yield (PGY), yield components, and flag leaf characteristics such as area, canopy temperature, or rolling index were evaluated. Data from previous studies of root traits recorded on the same set of genotypes at seedling and adult growth stages were used to search for possible associations with grain yield and other agronomic traits measured in the current work.

Results

Genotypic variation was found for all traits measured under both conditions. Grain yield and aerial biomass were reduced by 76% (from 5.28 to 1.97 Mg ha−1) and 66% (from 15.94 to 3.80 Mg ha−1), respectively in the dry year, whereas the harvest index increased by 32%. The breeding history of the germplasm (cultivar vs. landrace) had a significant effect on the traits studied. Landraces showed higher biomass only under drought (4.27 vs. 3.63 Mg ha−1), whereas modern cultivars out-yielded landraces only under non-drought conditions (5.56 vs. 4.49 Mg ha−1). Promising associations were found between root and agronomic traits, especially with grain yield, indicating that a profuse (large root length) and shallow (wide root angle) root system was related to increased yield of modern cultivars only in the dry year, without penalizing yield in the wet year.

Conclusion

Breeding programs could improve grain yield under Algerian, semi-arid conditions, by making crosses between selected landraces with good growth potential under drought and modern cultivars, with high efficiency of biomass conversion into grain, and searching for lines with acceptable agronomic performance, which combine these desirable traits from landraces and modern cultivars, with the presence of shallow and profuse root systems.

Introduction

Durum wheat (Triticum durum Desf.) is widely grown in the Mediterranean region, which represents 75% of the worldwide crop cultivation area (Graziani et al., 2014). Grain yields of durum wheat in Algeria are low, even compared to neighboring countries (Annicchiarico et al., 2002; Bahlouli et al., 2005). One of the main factors limiting yield is water stress, imposed by insufficient rain and its irregular seasonal distribution (Annicchiarico et al., 2005; Chennafi et al., 2006; Mekhlouf et al., 2006; Kourat, Smadhi & Madani, 2022; Benhizia et al., 2024), contributing to high year-to-year variability across the country (Bahlouli et al., 2005; Kirouani A. Boukhalfoune, Ouldkhiar & Bouzerzour, 2023), ranging from 1.8 to 3.6 tonnes per hectare (Bahlouli et al., 2005).

Wheat genetic resources, including landrace varieties, synthetic cultivars, and wild relatives are potential sources of alleles for enhancing drought tolerance and improving yield and its component traits (Moeller, Evers & Rebetzke, 2014; Cossani & Reynolds, 2015; Merchuk-Ovnat et al., 2016; Reynolds et al., 2017; Salgotra, Thompson & Chauhan, 2022). Wheat research has had some success in increasing drought and heat tolerance through the incorporation of genes from landraces, developing superior genotypes for cultivation in arid and semi-arid environments (Lopes & Reynolds, 2012; Cossani & Reynolds, 2015; Mondal et al., 2016; Crespo-Herrera et al., 2018). While breeding directly for grain yield under drought is feasible, it is hampered by the low heritability of the trait due to the presence of large genotype-by-environment interaction (GEI). This interaction of grain yield in wheat is a complex trait influenced by several agronomic, morphological, and physiological traits. These have been widely explored in wheat improvement programs to accelerate cultivar development (del Moral et al., 2003). If these traits are well associated with grain yield they could be used as indirect selection criteria (Gao et al., 2017; Demirel et al., 2023). Lopes & Reynolds (2012) further propose that genetic progress in yield is achievable through co-selection of several stress-tolerant traits associated with improved agronomic and physiological performance, all within a single variety. Some traits proposed for this purpose are early flowering and maturity (Chen et al., 2016; Mondal et al., 2016), increased harvest index (Royo et al., 2007), biomass (Xiao et al., 2012; Bustos et al., 2013; Gao et al., 2017), increased thousand kernel weight (Zhou et al., 2007; Tian et al., 2011; Lopes & Reynolds, 2012), high number of grains per spike (Yu et al., 2014; Würschum et al., 2018), morphological flag leaf features (Fan et al., 2015; Liu et al., 2018), and reduced canopy temperature differential (Lopes & Reynolds, 2010; Lopes & Reynolds, 2012; Gao et al., 2017).

In the last decades, the root system has received more attention for its potential contribution to drought adaptation (Bengough et al., 2004; Friedli et al., 2019; Asadullah et al., 2024). The importance of root systems for improving grain yield under drought-prone environments has been linked to traits like root depth (Lopes & Reynolds, 2010; Manschadi et al., 2010), root growth angle (El Hassouni et al., 2018), and distribution of root length density (Wasson et al., 2012). However, directly selecting for root traits in breeding programs faces significant challenges, primarily related to the cost and time required to evaluate the root systems of a large number of plants (Mace et al., 2012). Therefore, identifying reliable relationships between above-ground traits of interest and root traits, particularly during the early stages of plant growth, could be an efficient strategy to indirectly incorporate root traits into breeding programs (Bai, Liang & Hawkesford, 2013; Richard et al., 2018; Urbanavičiūte, Bonfiglioli & Pagnotta, 2022). Despite the huge effort put into root studies over the last decades, there is a knowledge gap on the impact of specific root characteristics on field performance of crops. The current study attempts to contribute to fill this gap, particularly for durum wheat grown under Algerian conditions, by linking the field performance of a set of diverse durum wheat genotypes with root characteristics of the same genotypes, previously studied with two different phenotyping systems proposed as breeding tools (Boudiar et al., 2020b; Boudiar et al., 2021). Specifically, the current study aims at (i) quantifying the phenotypic plasticity of a set of traits in response to drought and their relation to grain yield, (ii) determining how landraces and cultivars behave under contrasting rainfall conditions, and (iii) searching for useful associations between grain yield and other agronomic traits assessed in the current work with root traits assessed in previous studies.

Materials & Methods

Root data

Data on root characteristics were obtained from two separate experiments, previously published, including the same genotypes tested in the current work. The first experiment tested seven days’ old seedlings in a rhizoslide system, at 22 °C/18 °C in a growth chamber with a 12 h light/darkness photoperiod. Traits related to root system architecture and growth were measured (Boudiar et al., 2020b). In the second experiment, the same genotypes were grown in field conditions until 10 days after anthesis. Then, a shovelomics method was followed to extract the topsoil root system (to a depth of 20 cm) and various root traits were obtained such as traits related to length, root system width and root density (Boudiar et al., 2021). These two experimental setups have been proposed as pre-breeding or breeding methods to screen for desired root features (please, see the references mentioned in the two original articles).

Experimental site

Field experiments were conducted over two consecutive rainfed cropping seasons (2016/2017 and 2017/2018) at the experimental station of the Field Crops Institute (ITGC), Sétif, Algeria, located in the eastern highlands of Algeria. This site is representative of a large agricultural semi-arid region where the dominant farming system is based on cereal and sheep production (Bahlouli et al., 2005). The experimental site is in a Mediterranean semi-arid region. Most precipitations occur during the cold season, whereas the summer is hot and dry (Baldy, 1986; Chennafi, 2012). The cumulative rainfall of 369.3 mm is considered a baseline below which the year is considered dry (Chennafi, 2016). The soil texture is a sandy loam, with a water holding capacity of 26 mm, a pH of 8.4 and electric conductivity of 0.13 mS/cm. The soil is rich in organic matter (18 g/kg) and contains 0.08 g/kg nitrogen, 0.17 g/kg phosphorus (P2O5), 0.23 g/kg potassium (K2O), 0.36 g/kg manganese, 14.3 g/kg calcium oxide (CaO) and 0.02 g/kg sodium. The field trials during the dry and wet cropping seasons were conducted under rainfed conditions, without irrigation.

Plant material and experimental conditions

Thirty durum wheat varieties currently or historically grown in Algeria were evaluated. This diverse set included local landraces and modern cultivars from different geographical origins (Table S1), to gauge the effect of modern breeding on semi-arid field performance. Most of these genotypes were previously assessed for root traits at seedling (Boudiar et al., 2020b) and adult plant stages (Boudiar et al., 2021). Those data indicated the presence of root architecture variation, partially related to plant breeding history, and thus this set was considered as a good test material to link agronomic performance measured in this study to root traits.

Sowings were done on November 28, 2016, and December 11, 2017, with a sowing density of 300 seeds m−2. The trial was arranged in a randomized complete block design with three replications (complete blocks). Plots were 2.5 m long, and 1.2 m wide, resulting in plots of three m2 (6 rows, 20 cm apart). Fertilization was applied twice: first, triple superphosphate (0.46.0) was applied during soil preparation, at a rate of 45 kg ha−1, and later, at tillering stage, urea (46%) was supplied, at a rate of 100 kg ha−1. Weeds were controlled with the dicotyledonous plants’ herbicide “GRANSTAR” (tribenuron-methyl) applied at the jointing stage.

Trait measurement

The following traits were recorded in the two experiments, on a plot basis, or in samples, as indicated:

• Plant emergence (PEM) was estimated by counting the number of plants, in a row segment of 1.4 m (first year) or two m (second year), in the first week of February, and then converted to plants m−2.

• Days to heading (DTH) was recorded as the number of days since sowing until the date when 50% of the spikes were halfway out of the flag leaf sheath (stage Z55).

• Plant height (PH, cm), from the soil to the tip of spikes (awns excluded), the average height of 10 random plants per plot was recorded at maturity.

• Plot grain yield (PGY), was the weight of the combined harvested plot, converted to Mg ha−1, after adding the weight of the hand-harvested sample (see below).

The following traits were recorded as indicated, at specific time points:

• Leaf rolling at noon (LRN) was scored on a single date for all genotypes, around noon, as described in Bellon & Revees (2002), at heading stage, using a visual scale from 1 to 5 (1 = no-leaf rolling, 2 = leaf rim starts to roll, 3 = leaf is shaped like a V, 4 = rolled leaf rim covers part of the leaf blade, 5 = leaf is completely rolled).

• Canopy temperature (CT, ∘C) was recorded on the same day at noon, when all genotypes reached heading stage, by using an infrared thermometer (Model FLUKE 62 MAX) pointing to a healthy flag leaf.

• Flag leaf area (FLA, cm2) was assessed on a single day for all genotypes, towards the end of the heading stage, based on ten flag leaves randomly sampled from each plot. The mean length (FLL, cm) and width (FLW, cm) of 10 flag leaves were measured with a ruler. Then, the leaf area was calculated by the following formula (Zeuli & Qualset, 1990): FLA=FLL×FLW×0.749.

• Relative water content (RWC, %). One flag leaf per plot was sampled, immediately placed into pre-weighed and airtight vials, and transferred to the laboratory, where the fresh weight (FLFW, mg) was recorded. Then, leaves were submerged for four hours into distilled water. Leaves were then blot-dried with filter paper and weighted, to obtain the turgid weight (TW, mg). The samples were then oven dried at 80 °C for 72 h to obtain the dry weight (FLDW, mg), and the following formula (Barrs & Weatherly, 1962) was used to calculate RWC: RWC%=FLFW−FLDWTW−FLDW.

• Specific flag leaf weight (SLW, mg cm−2) was calculated according to the following formula: SLW=leafdryweight/leafarea.

All the following traits were estimated based on an above-ground biomass sample harvested at full maturity, from segments of one representative row, of 1.4 m or two m length for the 2016/2017 and 2017/2018 cropping seasons, respectively:

• The number of spikes per sample and the total number of grains per sample were recorded. From them, the number of grains per spike (GNS), spikes per area (SNA), and grain number per area (GNA) were calculated.

• Above-ground biomass (AGB, Mg ha−1).

• Harvest index (HI): was derived as 100 times the ratio of grain yield to total biomass.

• Thousand-kernel weight (TKW, g), estimated based on the weight of 250 grains.

Data analyses

Analyses of variance (ANOVA) were performed considering genotype, type (landrace vs. cultivar), year and their interaction as fixed effects, and block nested within year as a random effect. Contrasts within the ANOVA analyses were used to compare cultivars vs. landraces. The variation of genotype within type and its interaction with year was deduced from those of genotype and type. Block within year was used as the blocking factor; therefore, the interaction block × year (first residual) was used to test the significant effect of year on the measured traits, while the rest of the interactions, including the block factor (second residual), were used to test the significant effect of other factors. Multiple means comparisons were carried out using an LSD test, at 0.05 level of significance. All these statistical analyses were performed using Genstat software, version 18 (Payne et al., 2009). Radar diagrams were created using the fmsb R package (Nakazawa & Nakazawa, 2019), and a principal component analysis was performed using the FactoMiner R package (Lê, Josse & Husson, 2008).

Results

Growing conditions

The temperatures registered during the two cropping seasons were typical of the region, with warm autumns, cold winters, and warm to hot springs. The average temperature for the season in 2016/2017 (13.39 °C) was higher than that recorded in 2017/2018 (11.9 °C), for every month, except January (Fig. 1).

Figure 1 Accumulated monthly rainfall and mean monthly temperature recorded during the cropping seasons 2016/2017 and 2017/2018 referred to as drought and wet year, respectively.

Regarding precipitation, the 2017/2018 season presented more accumulated rainfall than that of 2016/2017 (440.7 mm vs. 195.12 mm, respectively). These aggregated rainfalls were clearly higher and lower, respectively, than the historical mean of the region for the cropping season, 315.7 mm (Wikipedia, 2025). The largest differences between the two years were observed during the months of most active vegetative and reproductive growth (March, April, and May, with 90.4, 75.4, and 42.7 mm of difference, respectively).

According to the differences in rainfall and temperature between seasons, hereinafter 2016/2017 and 2017/2018 will be referred to as dry/unfavorable and wet/favorable years, respectively, unless stated otherwise. This circumstance allows for a comparison of the performance of the genotypes in two highly contrasting conditions regarding water availability.

Genotypic variation and responses to drought

To reduce redundancy caused by multicollinearity among some traits and to simplify result interpretation, the initial set of 23 assessed and calculated traits (Supplementary File S1) was condensed to a more representative subset using cluster analysis. Agronomic relevance was also a key criterion for selecting traits for further analysis (Fig. S1).

The analyses of variance revealed significant genotypic differences for most traits across both cropping seasons (Table 1). Under drought conditions, the two groups (modern cultivars, and landraces) behaved differently for several traits, mainly GNS, AGB, HI, and FLA. Within each group (landraces and modern cultivars), genotypic differences were observed for the same traits identified in the entire dataset.

Table 1 Analyses of variance of assessed traits under drought and wet year.

	Drought year	Wet year	
Traits	Genotype	Type	Genotype within type	Genotype	Type	Genotype within type	
PEM	1,621*	1,267	1,634*	1,769*	45	1,830*	
PH (cm)	255***	3,697***	132***	931***	21,358***	201***	
SNA	2,122*	1,010	2,161*	5,144	39,376**	3,921	
GNS	22.38*	0.57*	23,20*	51.31	5.67	52.90	
GNA	1,684,801	493,430	1,727,350	6,963,736	44,728,456*	5,614,996	
DTH (days)	43.4	777.8	17.2	46.0***	795.8***	19.3***	
TKW (g)	32.28***	8.15	33.10***	45.64***	8.81	47.00***	
AGB (Mg ha−1)	0.79	7.54***	0.55	12.35	5.67	12.58	
PGY (Mg ha−1)	0.15	0.35	0.15	2.36***	20.11***	1.73**	
HI	0.04	0.30**	0.03	0.01***	0.11***	0.01**	
FLA (cm2)	220,630***	2,569,805***	136,730***	296,656***	1,589,026***	250,499***	
SLW (mg cm−2)	6.3E-5*	2.0E-4*	6E-5*	5.18E-5	1.97E-4*	4.7E-5	
LRN	2.34***	4.63***	2.30***	2.29***	4.33***	2.22***	
RWC (%)	16.63	164.02***	11.40	24.15	5.82	24.80	
CT (°C)	4.43	9.63	4.25	4.87	0.06	5.05	
	Combined ANOVA	
	Genotype (G)	Year (Y)	Type (T)	Genotype within type	Genotype × Year	Type × Year	G within T × Y	
PEM	1,574*	17,303	416	1,616*	1,815*	896	1,848*	
PH (cm)	986***	49,361***	21,414***	256***	200***	3,641***	77***	
SNA	4,408*	344,525*	26,499**	3619,32	2,858	13,888*	2464,07	
GNS	48.01**	3,299.68***	4.91	49,54**	25.68	1.33	26,55	
GNA	4.938E+06*	9.253E+08**	3.305E+07**	3,933,928	3.020E+06	2.261E+07**	2,320,357	
DTH (days)	79.11***	7,588.20***	1,573.50***	25,74***	10.37***	0.051	10.74***	
TKW (g)	61.65***	15,263.73***	0.006	63,86***	16.20***	16.96	16.18***	
AGB (Mg ha−1)	7.07	6,632.60***	13.14	6.86	6.07	0.06	6.28	
PGY (Mg ha−1)	1.49***	546.39***	12.87***	1.09***	1.03***	7.58***	0.79***	
HI	0.038**	1.10**	0.39***	0.03	0.20	0.02	0.02	
FLA (cm2)	401,060***	16,798,539**	4,100,184***	268,948***	116,225***	58,651	118,280***	
SLW (mg cm−2)	7.43E-5**	8.17E-3***	3.97E-4***	6.3E-5*	4.8E-5	1.6E-7	4.3E-5	
LRN	4.63***	0.01	8.96***	4.48***	0.01	0.00	0.01	
RWC (%)	14.42	3,523.21***	54.02	13.00	26.37*	115.82**	23.17	
CT (°C)	5.39	282.93**	4.11	5,43	3.92	5.59	3.86	
Notes.

PEM, Plant emergence per area; PH, Plant height; SNA, Spike number per area; GNS, Grain number per spike; GNA, Grain number per area; DTH, Days to heading; TKW, Thousand kernel weight; AGB, Above ground biomass; PGY, Plot grain yield; HI, Harvest index; FLA, Flag leaf area; SLW, Specific leaf weight; LRN, Leaf rolling at noon; RWC, Relative water content; CT, Canopy temperature

* Indicate significant differences at 0.05 level of significance.

** Indicate significant differences at 0.01 level of significance.

*** Indicate significant differences at <0.001 level of significance.

Mean square values without asterisks: no significant difference at 0.05 level of significance.

The significant differences between cultivar and landrace groups observed under drought, for GNS (19.9 vs 19.6), AGB (3.63 vs 4.27 Mg ha−1), and RWC (80.0 vs. 83.1%), were not observed in the wet year, and the opposite was true for SNA (331.1 vs 281), GNA (9,529 vs 7,878), DTH (120.7 vs 127.4 days), and PGY (5.56 vs. 4.49 Mg ha−1), indicating specific responses to stress of the two germplasm pools. RWC showed a significant positive correlation with AGB (0.423) under drought, but this did relationship did not carry over to grain yield. This occurred due to the combination of lower HI and higher RWC of landraces under drought. In the wet year, genotypic differences within each group (landraces and modern cultivars) were observed for all traits that showed significance in the dry year, except for SNA, GNS, DTH, PGY, and SLW (Table 1). The combined ANOVA of the two cropping seasons highlighted the strong effect of the year on all traits, except for plant emergence (PEM) and LRN. A significant interaction between genotypes and year was detected for PEM, PH, DTH, TKW, PGY, FLA and RWC. Genotypes and genotypes within type behaved similarly across years for all traits except for RWC (Table 1).

In the dry year, a drastic reduction of values for most traits was observed, except for PEM (no change) and for HI and SLW, which increased significantly by 31.8% and 19.5%, respectively, compared to the wet year (Fig. 2A, Table S2). Grain yield and above-ground biomass (AGB) were the traits most affected by drought (reductions of 66.1 and 76.1%, respectively) followed by the number of grains (49.8%) and TKW (34.5%), with traits recorded on the flag leaf being the least affected ones. No drought effect was observed for LRN. In the dry year, genotypes were earlier overall (DTH) by about 13 days (Table S2) compared to the wet year, probably due to the faster accumulation of growing degree days due to the higher temperatures of the season. During the wet year, most traits exhibited a wider range of variation compared to drought conditions. However, harvest index (HI) displayed a greater range of variation under drought stress, suggesting differential genotypic sensitivities to stress during grain filling (Table S2).

Figure 2 Radar diagram, comparison between drought (orange) vs. wet year (green) (A) and cultivars (blue) vs. landraces (yellow) under drought (B) and wet year (C) for traits recorded at field.

Underlined traits indicate significant difference between different groups at 0.05 significance level. PEM, Plant emergence per area; CT, Canopy temperature; LRN, Leaf rolling at noon; RWC, Relative water content; SLW, Specific leaf weight; FLA, Flag leaf area; PGY, Plot grain yield; HI, Harvest index; AGB, Above ground biomass; TKW, Thousand kernel weight; DTH, Days to heading; GNA, Grain number per area; GNS, Grain number per spike; SNA, Spike number per area; PH, Plant height.

Cultivars vs. landraces under contrasting water conditions

Across the two seasons, landraces and modern cultivars exhibited distinct behaviors. Landraces consistently displayed greater PH, FLA, and leaf rolling (LRN) compared to cultivars in both years. Conversely, cultivars generally had higher HI and SLW. Responses for other traits varied by germplasm type and season. Landraces maintained higher above-ground biomass (AGB) and relative water content (RWC) under drought stress, but this advantage was not observed in the wet year. Interestingly, spike number per area (SNA), grain number per area (GNA), and PGY were higher for cultivars under wet conditions, but these values became similar for both types under drought (Table 2, Figs. 2B, 2C).

Table 2 Comparison of means for the landrace and cultivar groups (type effect) for the assessed traits across drought and wet year.

	Drought year	Wet year	
Traits	Cultivars	Landraces	Cultivars	Landraces	
PEM	207a	197a	183.9a	186.8a	
PH (cm)	61.1b	75.5a	88.7b	123.5a	
SNA	233a	225a	331a	281b	
GNS	19.9a	19.6b	28.7a	28.1a	
GNA	4,604a	4,417a	9,529a	7,878b	
DTH (days)	107.7a	114.4a	120.7b	127.4a	
TKW (g)	34.80a	35.55a	53.82a	52.32a	
AGB (Mg ha−1)	3.63b	4.27a	15.86a	16.17a	
PGY (Mg ha−1)	1.83a	1.69a	5.56a	4.49b	
HI	0.43a	0.37b	0.34a	0.26b	
FLA (cm2)	13.73b	17.57a	19.97b	23.08a	
SLW (mg cm−2)	8.39a	8.04b	7.06a	6.70b	
LRN	2.71b	3.21a	2.71b	3.21a	
RWC (%)	80.0b	83.1a	89.9a	89.1a	
CT (°C)	26.7a	25.97a	28.9a	29.0a	
Notes.

PEM, Plant emergence per area; PH, Plant height; SNA, Spike number per area; GNS, Grain number per spike; GNA, Grain number per area; DTH, Days to heading; TKW, Thousand kernel weight; AGB, Above ground biomass; PGY, Plot grain yield; HI, Harvest index; FLA, Flag leaf area; SLW, Specific leaf weight; LRN, Leaf rolling at noon; RWC, Relative water content; CT, Canopy temperature. Different letters indicate that mean of cultivars and landraces were significantly different at 0.05 level of significance.

While landraces experienced a smaller reduction in biomass production during drought, their inherently lower HI limited their ability to convert biomass into grain yield. Consequently, cultivars yielded slightly lower than landraces under drought, but the difference was not statistically significant (Table 2, Fig. 3). Differences in PGY between types were primarily driven by variations in SNA and GNA, not by thousand kernel weight (TKW), which showed no significant difference between groups under either condition (Fig. S2). Favorable grain-filling conditions in both years likely benefitted both landraces and cultivars. Therefore, yield differences were related to different efficiencies in the formation of yield-bearing structures before grain filling (spikes and grains). Grain yield (PGY) of individual genotypes within each group displayed variation under both conditions (Table 1, Table S3), but statistically significant differences were only observed in the wet year.

Figure 3 Above ground biomass (AGB) and plot grain yield (GY) of cultivar and landrace groups across drought and wet years.

Bars represent standard error of the means (±SEM).

Relationships between traits among years

Under drought conditions, GY showed a stronger association with GNA and HI compared to traits related to biomass (AGB, PH, and FLA). Additionally, the first two components of the PCA analysis, which explained 30.25% (PC1) and 19.70% (PC2) of the total variability, indicated that traits like LRN, RWC, and TKW (Fig. 4A) displayed negative correlations with yield under drought. This negative correlation is likely caused by the contrasting responses observed between landraces and modern cultivars, as reflected in the first principal component of the PCA analysis.

Figure 4 Principal component analyses of 30 durum wheat, cultivars (blue) and landraces (yellow) based on field traits evaluated under drought (A) and wet year (B).

PEM, Plant emergence per area; CT, Canopy temperature; LRN, Leaf rolling at noon; RWC, Relative water content; SLW, Specific leaf weight; FLA, Flag leaf area; PGY, Plot grain yield; HI, Harvest index; AGB, Above ground biomass; TKW, Thousand kernel weight; DTH, Days to heading; GNA, Grain number per area; GNS, Grain number per spike; SNA, Spike number per area; PH, Plant height.

In the wet year, the PCA with PC1 and PC2 explained a total variability of 45.13, where 27.90 and 17.23% were attributed to PC1 and PC2, respectively. The grain yield had a very large negative loading in the first principal component, reflecting the higher GNA and PGY of modern cultivars under those conditions (almost all with negative loadings). The second component was less significant that year (explaining 17.2% of the variation), indicating less variation beyond the type contrast under wet conditions (Fig. 4B). However, in the dry year, grain yield had relatively large loadings in both the first and second principal components (Fig. 4A). This suggests a wider distribution of yield-related variation within both modern cultivars and landraces under drought. When correlations were analyzed between corresponding traits measured in both dry and wet seasons, some significant correlations can be highlighted. These included DTH (0.77), PH (0.80), and TKW (0.59) (Table S4). PGY was equally influenced by GNA and HI in both years, but SNA (number of spikes per area) was important in the wet year, and GNS was determinant only in the dry year, hinting at a role of floret survival under drought as a tolerance mechanism. Another set of revealing correlation coefficients was that of RWC with other morpho-physiological traits and biomass. In the wet year, all RWC correlations were non-significant. However, in the dry year, the genotype by environment interaction (type x year) detected for RWC, was determinant in the presence of positive correlation of RWC with biomass (AGB), and for the significant correlations of RWC with LRN, CT, FLA. Apparently, higher RWC was associated with lesser canopy temperature, larger flag leaf and higher leaf rolling, only under drought.

Root features and agronomic association depended on water availability

Correlations between root traits measured in previous studies on seedlings (Boudiar et al., 2020b) and adult plants (Boudiar et al., 2021) (Supplementary File S2) for the same set of cultivars revealed interesting relationships with the agronomic traits evaluated under contrasting conditions in the current work. The previously reported differences between modern cultivars and landraces (Boudiar et al., 2020b; Boudiar et al., 2021) were also observed in this study. To focus on the potential influence of root traits on agronomic performance beyond genotypic differences between landraces and modern cultivars, we calculated correlations only for the subset of modern cultivars (n = 22). The seedling root-opening angle (RoA) was positively associated with grain yield and HI in the dry year, but not in the wet one. This indicates that a wider seminal root angle would be advantageous only under drought conditions, without penalizing yield in the wet year (Figs. 5, 6, Table S5).

Figure 5 Relationships of root growth opening angle (seedling), total projected structure length and median structure width (adult plant) with grain yield under drought and wet year conditions.

Figure 6 Relationships of root growth opening angle (adult plant), total root length (seedling) and median structure width (adult plant) with harvest index under drought and wet year conditions.

Similar to the root opening angle, shorter seedling roots increased harvest index (HI) only under drought conditions. Conversely, for adult plants, longer total root length at a depth of 20 cm, measured as total projected structure length (tpSL), benefited plot grain yield (PGY) exclusively under drought. Interestingly, root systems with a higher median structure width (mSW)—a measure of how densely roots are packed together—were detrimental to grain yield and harvest index, but only under drought. This suggests that sparse root systems with a low mSW are more advantageous during drought stress.

In addition, other noteworthy correlations emerged between root traits and agronomic performance (Table S5). For example, a longer primary root length at the seedling stage was negatively associated with HI and grain number per spike (GNS) only in the wet year, with no impact under drought. Seedlings with a higher number of roots displayed increased relative water content (RWC) and lower shoot dry weight (CT)—but again, only under drought stress (Table S5). In the wet year, seedling root length presented positive correlations with PH and SNA, root number of seedlings was positively associated with PH and LRN. Root dry weight of adult plants presented a negative correlation coefficient with aerial biomass and positive ones with number of emerged plants and grain number per spike (Table S5).

Discussion

Effect of drought on agronomic traits

The rain pattern observed during this experiment, with drought conditions occurring around the flowering stage, is typical of Algerian cereal-growing regions (Annicchiarico et al., 2005; Mekhlouf et al., 2006). The eastern high plateaus of Algeria experience variable rainfall conditions. Furthermore, it is predicted that temperatures will significantly increase and rainfall will significantly decrease during the twenty-first century (Chetioui & Bouregaa, 2024). The rainfall accumulated during the dry and wet years (195.1 and 440.7 mm) was close to the extremes of the range of precipitations between 168.7 and 517.3 mm, found by Chennafi et al. (2006), and also to the rainfall distribution found for the last 20 years (2005–2024), from 163.0 to 444.2 mm, according to http://www.infoclimat.fr. Accordingly, in the current study, the grain average yield achieved in the dry year (1.79 Mg ha−1) fell slightly below the Algerian average wheat yield (1.9 Mg ha−1) (FAO, 2018), contrary to the wet year, in which grain yield (5.28 Mg ha−1) was an extraordinary year, well above the historic mean. In fact, Bahlouli et al. (2005) reported yields ranging from 1.79 to 3.58 Mg ha−1 during six successive cropping seasons (1997/1998 to 2002/2003) in the same region where our experiment was conducted. The two field seasons were highly contrasting in terms of amount and distribution of rainfall received seasons, cumulative. This circumstance allowed a meaningful comparison of the genotypes, despite being tested in just two seasons. The dry year in our study experienced extended drought stress. Rainfall data (particularly the high difference in precipitation between years during March-May) and the responses of above-ground biomass and thousand kernel weight (TKW) in the dry year point to this conclusion. Grain yield and biomass showed significant reductions (66% and 76%, respectively) under drought. Conversely, harvest index displayed an opposite trend, actually increasing by 31%.

These combined results suggest severe water stress during the critical period of maximum biomass accumulation in the dry year. This could have limited the source of resources available for grain filling. The lower TKW under drought indicates that water stress likely affected the pre- and post-heading period, including grain filling. However, the higher harvest index under drought suggests a relatively balanced reduction in source (biomass) and sink (grain) potential. In other words, grain filling itself was not as severely restricted by drought compared to the source limitation. The excellent biomass accumulation and lack of severe limitations during grain filling in the wet year suggest a potential for improving durum wheat yields under favorable conditions, as indicated by the low harvest index observed. The reduction of grain yield in the dry year was a consequence of a rather similar reduction in all yield components: 26% less spikes per unit area, 29% less grains per spike, and 34% less TKW. This fact indicates that drought effects were probably complex, affecting different features throughout the crop’s life cycle. The higher HI attained in the dry year indicates a potential for this trait that was not achieved in the wet year, suggesting room for HI improvement in the crop. Our findings are consistent with those of Bidinger, Musgrave & Fischer (1977) regarding the impact of drought on yield components. However, they differ from the results reported by Aggarwal et al. (1986) and Giunta, Motzo & Deidda (1993) on the specific traits most affected by drought. As Lopes & Reynolds (2012) and Zhang et al. (2018) point out, the traits that contribute most to higher grain yields can vary depending on the specific environmental conditions (e.g., drought severity, timing, and duration). The contrasting findings from these studies likely reflect these variations in drought scenarios.

Cultivars vs. landraces

Previous studies report conflicting findings on biomass production, with some showing higher biomass in landraces (Ayadi et al., 2015; Carranza-Gallego et al., 2019; Eser, Soylu & Ozkan, 2024) and others favoring modern cultivars (De Vita et al., 2007; Royo et al., 2007). Our study agrees with Siddique et al. (1989), who also found higher biomass productions of landraces only under drought conditions. Our landraces showed higher RWC % only in the dry year, which could indicate that landraces had the ability of maintaining better leaf water status and growing more (yielding higher biomass) than cultivars under water limited conditions. According to Chaouachi et al. (2023), this trait (RWC) was reported to be involved in drought tolerance in durum wheat. Under favorable conditions, no significant difference was detected. While high biomass at anthesis can minimize yield reduction under drought stress (Simane, Peacock & Struik, 1993; Bahlouli et al., 2005), the difference in grain yield between landraces and cultivars wasn’t significant under drought in our study. The higher number of spikes and grains per area observed in cultivars compared to landraces in the wet year likely explains their superior grain yield. This difference could be due to variations in tillering ability (as observed in the same germplasm by Boudiar et al., 2021), production of barren tillers, or floret fertility (Royo et al., 2007). Grain number per spike was similarly reduced for both groups, making floret fertility the least likely explanation. Since we did not directly measure unfertile tillers, we can infer they were more abundant in landraces, based on our previous results carried out with the same genotypes (Boudiar et al., 2021). This would explain their inherently lower harvest index, considering the lack of differences in grains per spike and TKW between the groups. Therefore, the landraces tested failed to convert their abundant biomass into high grain yield under either condition, likely due to a combination of lower tillering ability and a higher presence of tillers with fewer spikes.

Besides these general observations, we found significant genotype-by-environment interaction for grain yield, not just between landraces and cultivars but also within each group. Not all landraces, nor modern cultivars, behaved similarly. Some landraces, like Oued Znatie 368 (ranked 27th in yield under wet conditions but 9th under drought), performed relatively better under drought. It was highly drought tolerant across the entire cycle, as shown by small reductions for all yield components under drought. This landrace was also the best in maintaining a high RWC in the dry year, suggesting a good water status that would favour biomass production. It is interesting to note that the correlation coefficient of RWC with AGB was negligible in the wet year, but increased up to 0.423 in the dry year (P = 0.01). This relationship points at a role of maintaining shoot water status in sustaining good biomass growth under drought. We could not identify any root traits related to the good relative performance of Oued Znatie 368 under stress. Other landraces, however, like Montpellier and Mohammed Ben Bachir exhibited relatively poorer performance under drought stress than in the wet year. Contrasting responses were also observed among modern cultivars. Ofanto and Simeto were the most drought-sensitive, while Oued El Berd displayed the highest drought tolerance. Modern cultivars Mexicali 75 and Megress were the most stable performers across conditions, while Cirta and GTA Dur were slightly less stable but still yielded well overall. Our findings suggest that landraces produced more biomass than modern cultivars under drought, indicating a better growth ability and physiological adaptation mechanisms—supported by their higher relative water content and flag leaf area in harsh conditions. However, they were less efficient than cultivars at converting biomass into grain yield. This trait was likely favored during modern breeding programs. Over the two years, modern cultivars clearly out-yielded landraces. If the environments tested were equally representative, it would be indicative of an overall superiority of modern cultivars. We can expect that seasons like the dry year are more likely to occur in Algeria, given the long-term data on wheat yield averages, which never surpassed 2,000 kg ha−1. Even if a season like the wet one occurred once every ten years, the averages of modern cultivars Mexicali 75, Sitifis, GTA Dur and others would still surpass those of the landraces. Therefore, selected landraces like Oued Znatie 368 could be a valuable genetic resource to improve the vegetative growth ability of modern cultivars under drought, especially when combined with the improved harvest index of modern materials. The combination of these two traits may be difficult, due to the possible trade-offs between high vegetative growth and optimum grain filling. This is hinted by the correlations of RWC with biomass and harvest index under drought. One could speculate that mechanisms favoring maintenance of shoot water status (high RWC) may have a negative effect in favoring the retranslocation of assimilates to the grain during grain filling, hence reducing drastically harvest index. Although we cannot conclude this from our data, this question opens avenues of future research. Additionally, high vegetative growth may reduce available soil water beyond acceptable limits for good grain filling. However, there may be enough diversity to optimize the two traits in a well-balanced plant for Algerian environments. The Algerian landraces also have a number of unfavorable traits like lodging, late flowering, thus their integration into elite backgrounds should be made with caution. A sensible breeding strategy would be to develop introgression populations (BC1 or BC2) with Oued Znatie 368 and the best elite cultivars, search for QTL in segregating populations with field or controlled conditions phenotyping, and advance generations using marker assisted selection for yield traits and RWC, and against plant height, late flowering and lodging, combined with field testing.

Relationship of root traits and grain yield under contrasting water conditions

The expression of root traits varies depending on soil conditions and water availability (Tuberosa et al., 2002; Lynch, 2007; Gaur, Krishnamurthy & Kashiwagi, 2008; Bektas et al., 2023). This variability can lead to either positive or negative relationships with grain yield (Kell, 2011; Bishopp & Lynch, 2015; Schoppach et al., 2013). To effectively incorporate root traits into breeding programs, we need to identify which traits are most beneficial for yield improvement under specific conditions. Interestingly, our study found that relationships between root traits (measured at both seedling and adult stages) and field performance were only significant in the dry year. This finding aligns well with observations by Colombo et al. (2022) for durum wheat. The sustained drought stress experienced in the dry year likely strengthened the connections between root traits measured at seedling stage and agronomic performance. Fluctuating stress conditions over short periods are less likely to produce such clear relationships (Lynch, 2013). Another interesting observation is that root traits beneficial under drought had no impact on yield under wet conditions. The dependence of the relationship between root features and agronomic performance from the environment has also been documented in other studies (Roselló et al., 2019; Colombo et al., 2022). The differential root relationships among studies could be due to the sensitivity of roots to soil conditions and plant growth stage, as drought could shift the biomass allocation between shoots and roots (Bektas et al., 2023) or even among different types of roots (Boudiar et al., 2020a).

Our findings suggest that a well-developed but shallow root system with either fine or sparse roots (low mSW) is associated with increased grain yield in modern cultivars, only under sustained drought conditions. This root conformation may lead to more efficient capture of water from the upper soil layers, which are the only ones holding the water from scarce rain events. However, other soil conditions could be implicated as reported by Van der Bom et al. (2024): wide-angled genotypes developed more roots in the topsoil, where phosphorus was available, and produced greater biomass at anthesis under these conditions.

Our results, indicating a positive effect of large root length (tpSL) and low median structure width (mSW) on drought performance, could be explained by the advantage of a system with loose, sparse roots. This allows for efficient exploration of the entire root zone, in contrast to a vigorously growing root system where roots clump together, competing for resources and having less overall contact with the soil compared to their total volume. This explanation aligns with the findings of Figueroa-Bustos et al. (2020), who observed greater drought tolerance in wheat cultivars with smaller root systems compared to those with larger ones. It is important to note that the correlation between total projected structure length (tpSL) and mSW (Boudiar et al., 2021) is rather weak. This suggests that breeding programs could potentially select for favorable expressions of both traits within the same cultivar. The correlations we observed were conducted using only modern cultivars to avoid the confounding influence of differences between landraces and modern cultivars.

These findings suggest that breeding programs in Algeria, and potentially other Mediterranean environments, could target root traits for yield improvement. While our results, regarding the importance of a large root system, align with those of Roselló et al. (2019), they differ from previous studies that associated compact root systems with higher drought tolerance (Manschadi et al., 2006; El Hassouni et al., 2018). The value of these traits for breeding could be tested through the use of divergent selection for tpSL and mSW in segregating populations of contrasting parents, and wide field evaluation of the resulting divergent lines. Advantages and trade-offs of either rooting morphology would be revealed with this approach.

Conclusions

Modern cultivars outperformed landraces in grain yield under wet conditions, primarily due to a higher number of grains and spikes per unit area. Our study suggests that selecting for a shallow root system with a wide opening angle at the seedling stage that develops into a large and well-distributed root system in mature plants could be a valuable breeding strategy for improving grain yield under sustained drought, without sacrificing yield under favorable conditions.

Several cultivars, including Cirta, GTA Dur, Korifla, Massinissa, Megress, Mexicali 75, Sitifis, Polonicum, Vitron, and Waha, demonstrated good grain yield under both wet and dry conditions. These varieties are well-suited for semi-arid regions and likely possess valuable traits for adaptation to a wide range of environmental conditions. Oued Znatie 368 is the only landrace among those tested in this study, with distinctive drought tolerance traits that could be used for durum wheat breeding in Algeria.

A promising strategy for durum wheat breeding programs in Algeria and similar semi-arid regions, would be to combine the superior drought tolerance and vegetative growth of landraces with the efficient harvest index of modern cultivars (trade-offs allowing), along with the presence of shallow and well-developed root systems associated with increased yield under drought conditions. Promising yield-related traits like seedling root angle, tpSL, mSW, can be tested using medium-throughput phenotyping methods like rhizoslides or shovelomics, therefore well-suited to use in breeding programs.

Supplemental Information

Table S1 List of the 30 varieties of durum wheat evaluated during the two cropping seasons

Table S2 Minimum, maximum, mean comparison and drought reduction (Redu %) for the recorded traits

Table S3 Comparison of plot grain yield for drought and non-drought cropping season for 30 durum wheat varieties including cultivars and landraces

Comparison performed considering all genotypes together (cultivars and landraces).

Table S4 Pearson coefficients for the assessed traits under the cropping season 2016/2017 and 2017/2018, drought and wet year, respectively

Table S5 Pearson correlation coefficients, based on twenty-two modern cultivars, between root traits measured at seedling and adult plant growth stages and field traits in the dry and wet years

Figure S1 Process of selection of representative phenotypic traits through hierarchical clustering

The traits measured (initial traits) were subjected to cluster analysis for each season.

Figure S2 Comparison for grain number per area (GNA) and thousand kernel weight (TKW) of cultivars and landraces under drought and wet year

Data S1 Traits measured at field under drought (2017) and wet year (2018)

All the data points of each single trait per genotype per year, which were used for statistical analyses.

Data S2 Mean values of traits measured in rhizoslide, shovelomics and field experiment used to calculate the relationships among traits

The technical assistance in field trials provided by all the technical staff of Field Crops Institute (ITGC, Sétif, Algeria) is highly appreciated.

Additional Information and Declarations

Competing Interests

Author Contributions

Data Availability

The authors declare there are no competing interests.

Ridha Boudiar conceived and designed the experiments, performed the experiments, analyzed the data, prepared figures and/or tables, authored or reviewed drafts of the article, methodology; Data Curation, and approved the final draft.

Abdelhamid Mekhlouf conceived and designed the experiments, performed the experiments, authored or reviewed drafts of the article, supervision; Methodology, and approved the final draft.

Yacine Bekkar performed the experiments, prepared figures and/or tables, and approved the final draft.

Meriem Yessaadi performed the experiments, prepared figures and/or tables, and approved the final draft.

Adel Bachir conceived and designed the experiments, performed the experiments, authored or reviewed drafts of the article, and approved the final draft.

Larbi Karkour performed the experiments, prepared figures and/or tables, and approved the final draft.

Ana Maria Casas analyzed the data, prepared figures and/or tables, authored or reviewed drafts of the article, supervision; Resources, and approved the final draft.

Ernesto Igartua analyzed the data, prepared figures and/or tables, authored or reviewed drafts of the article, supervision; Resources; Methodology, and approved the final draft.

The following information was supplied regarding data availability:

The raw data are available in the Supplementary Files.

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
