# Peer review of "Enhancing drought resilience in durum wheat: effect of root architecture and genotypic performance in semi-arid rainfed regions"

_PeerJ, doi:10.7717/peerj.19096_

## Round 0.1 · original submission · Major Revisions

Dear Authors

The manuscript cannot be accepted for publication in its current form. It needs a major revision before publication. The authors are invited to revise the paper considering all the suggestions made by the reviewers. Please note that the requested changes are required for publication.

With Thanks

·

Basic reporting

Thank you for considering me for reviewing the manuscript “Performance of durum wheat under drought affected by root traits”. The manuscript investigates the agronomic and root traits of durum wheat genotypes under contrasting conditions in Algeria. The manuscript presents valuable research that aligns to improve crop resilience in semi-arid regions. The authors have successfully highlighted the importance of root system traits in drought adaptation and have provided meaningful results from field trials.

Suggestions:
Title:
The title could be improved to be “Enhancing drought resilience in durum wheat: Effect of root architecture and genotypic performance under rainfall conditions in semi-arid regions”

Abstract:
The abstract begins with a robust background, emphasizing the importance of developing drought-adapted genotypes for resilient agriculture in the Mediterranean region. The methods are briefly summarized, presenting the use of field trials over two years with contrasting rainfall patterns.
Key results are clearly presented, including the differences in grain yield and biomass reduction between drought and non-drought years. Also it could include more specific data points, such as actual grain yields in the dry and wet years. For instance: "Grain yield was reduced by 76% in the dry year, dropping to …kg/ha, compared to … kg/ ha in the wet year".
The mention of genotypic variation and the relationships between root traits and grain yield highlights important findings. The conclusion is concise and effectively summarizes the key finding: combining landraces' drought growth ability with modern cultivars' higher harvest index and specific root traits can improve yield under drought conditions.

Introduction:
The introduction effectively highlights the importance of durum wheat in the Mediterranean region, particularly under semi-arid conditions like those in Algeria, emphasizing the need for drought-tolerant genotypes, which is central to the study. It also provides a general background on the impact of drought stress on wheat production and briefly reviews previous research on wheat breeding and the challenges of improving drought tolerance.
However, the references need updating, as many are from before 2010, and there is a noticeable lack of recent studies from 2022-2024. Including more current references would strengthen the relevance of the literature review.
Additionally, the introduction does not clearly highlight the specific knowledge gap this research addresses. It is important to briefly explain how this study builds on prior work and fills existing gaps in understanding. Explicitly stating the knowledge gap, why this research is necessary, and what gap it aims to fill will enhance the study's novelty. For instance: "Despite significant advances in breeding for drought tolerance, the specific role of root system traits in enhancing durum wheat performance under drought conditions remains underexplored."

Methodology:
The methodology is well-detailed, but some sections need further clarification. The first paragraph could be divided into subsections, (i) Experimental sites and (ii) Plant material and experimental design. Each subsection needs more information, particulary the first needs more details on the experimental sites should be added, including its geographic coordinates (longitude, latitude, altitude) and soil properties (e.g., type, pH, nutrient content), meteorological data for the growing seasons, such as temperature fluctuations, rainfall distribution, and humidity levels,…..
The description of rainfall and temperature data in the results should be moved to the methodology section.
The root traits from previous studies are mentioned in a published paper, but should be briefly mentioned in Ms&Ms of this paper.

Results:
I suggest adding the supplementary Figure 3 to the main text; it is interesting and provides important results.
Table 2 displays only mean performance, not an analysis of variance.
Radar diagram is a good idea to compare drought and wet years and between cultivars and landraces under drought and wet conditions. However, the resolution of the Figure is currently inadequate leading to unclear and pixelated images. Figures 1-5 should be clear and legible for electronic publications at 100% zoom. Also, the figure caption should provide a clearer explanation of the abbreviated traits.

Discussion
While the authors have successfully demonstrated the positive correlation between root traits and grain yield under drought conditions, interpreting the negative correlations requires discussion. Including more literature-based reasoning or physiological explanations would strengthen this section. The discussion would benefit from further exploration of the practical implications of the findings. For example, how can the study’s findings be applied in breeding programs, and how can the promising root traits be incorporated into future breeding strategies for drought resilience in Mediterranean environments. The literature review is thorough, but newer references on root systems and drought adaptation might have been missed. Ensure the literature cited includes the most recent studies (especially after 2022).

Conclusion is well-written
Language: The manuscript is generally well-written, however, I suggest a thorough grammar and style revision to improve the flow of the text.
References: Please review and standardize the reference section according to style guidelines. Some details are missing, as page numbers in lines 448 and 458.

Experimental design

The experimental design is appropriate

Validity of the findings

The findings presented in the manuscript are valid, supported by robust data and sound statistical analysis.

Reviewer 2 ·

Basic reporting

The manuscript titled "Performance of Durum Wheat under Drought Affected by Root Traits," presents insights into how specific root traits contribute to drought resilience in durum wheat. However, several revisions are suggested to enhance the clarity and impact of the work.The language needs improvement, in terms of expression, grammar, and there are some areas where simplification could improve readability. Some sentences are long and could be broken up to improve readability.
In abstract, the statement that root data came from "previous studies" is vague. It leaves the reader uncertain about how directly comparable or relevant the root data is to the present study. The conclusion about improving grain yield through a combination of traits from landraces and modern cultivars lacks depth. It doesn't specify how these traits might be combined in a breeding program or whether trade-offs exist. Provide more insight into the challenges or considerations in combining these traits for breeding.
In introduction section, there is an overreliance on older studies (e.g., Bahlouli et al., 2005; del Moral et al., 2003), please integrate more recent research (within the last 5 years) to reflect current advancements in wheat breeding and phenotypic plasticity under drought stress. The text mentions several physiological traits like root growth angle, canopy temperature, and flag leaf features, but doesn't elaborate on why these specific traits are important in the context of durum wheat or their unique role in breeding programs. The connection between above-ground traits and root traits in drought tolerance is an interesting point but could be developed further to emphasise its novelty and importance in current wheat research.
In the Materials and Methods section, it is essential to provide more comprehensive details regarding the complete experimental set up e.g., number of blocks used, as well as a thorough description of the soil conditions as soil type, water holding capacity, nutrient status, and organic matter content. The inclusion of 30 durum wheat varieties is excellent for evaluating genetic diversity, but the study could benefit from more detailed explanations of the varietal selection process. Were these varieties chosen based on specific agronomic traits or drought tolerance traits? More transparency in selection criteria would add credibility to the study. Sowing density (300 seeds m²) and fertilizer applications are described, but more information on irrigation management during the trials is missing. Even in rainfed conditions, supplemental irrigation can sometimes be used, and it's critical to explicitly state how water availability was managed. The data collection protocols for traits such as plant emergence, days to heading, plant height, and grain yield are standard and reliable for wheat. However, it is unclear how leaf rolling and canopy temperature were standardized across plots. For instance, was canopy temperature recorded at the same time across blocks or was it adjusted for microclimatic variations? The leaf rolling and canopy temperature measurements provide useful insights into plant stress response, but the methodology for scoring leaf rolling (visual scale) may introduce subjectivity. Using more objective methods such as digital image analysis (e.g by ImageJ software) or physiological measurements would reduce potential bias. Principal component analysis (PCA) is an excellent approach for visualizing the relationships among traits, but the description here is somewhat brief. How were the components selected, and what proportion of variance did the first few components explain? The Materials and Methods section would benefit from a clearer separation between the field design (environmental conditions, varieties used) and the trait measurements (phenotypic and physiological parameters). Currently, the information regarding the experimental setup and trait measurements is somewhat mixed, which may lead to confusion.
The result section of the manuscript is written in a non-technical manner, which limits its utility for a scientific audience. There should be a strong focus on presenting the most promising results with a deeper and more technical analysis, emphasizing the statistical significance of key findings. In the growing conditions, the variation in temperatures and rainfall between the two cropping seasons provides a natural experiment for evaluating genotype performance under contrasting water conditions. However, the manuscript presents this information without linking it directly to the impact on specific traits or plant responses. In the genotypic variation and response to drought, significant genotypic differences were observed across traits such as grain number per spike (GNS), above-ground biomass (AGB), and flag leaf area (FLA). However, the description lacks detail on the magnitude of these differences and their statistical support. While the manuscript identifies a drastic reduction in grain yield (PGY) and biomass under drought (66.1% and 76.1%, respectively), these values should be framed within the context of statistical significance and variability across wet and dry years as well across various varities. Were the reductions consistent across all genotypes? In the trait correlations and root analysis, the potential for root traits (such as root angle and structure length) to enhance drought tolerance is intriguing. However, this section requires a more detailed description, especially considering the differences in correlations between wet and dry years.
In the Discussion section, it is important to provide a more in-depth exploration of the potential physiological mechanisms that may explain the superior growth performance of the landraces under drought conditions. Specifically, mechanisms such as the development of deeper rooting systems, enhanced osmotic adjustment capabilities, or more efficient water use strategies should be discussed in detail as potential adaptive traits. These physiological responses could offer insights into how the landraces maintain better growth and yield stability under water-limited environments. Furthermore, the current discussion primarily references studies from the late 1900s, which limits its relevance to contemporary scientific understanding. It is essential to place your findings in the context of more recent research, as significant advancements in drought physiology and plant adaptation mechanisms have been made in recent decades.
The conclusion regarding root traits is valid but could be made more specific. Mentioning how these traits might be integrated into breeding programs, along with specific genetic or agronomic approaches, would add more practical relevance. Provide more specific recommendations for incorporating root traits into breeding programs, including potential genetic markers or selection criteria for root traits under drought conditions. Mention whether the listed cultivars are already in commercial use or if further validation is needed before large-scale adoption. Acknowledge the study limitations and mention the future research directions. Mention any policy implications of the study.
Add full forms of all abbreviations for various traits in the footnotes of the figures. This will enhance clarity and improve the overall readability and accessibility of the figures. Remember that figures must be self-explanatory.
It is recommended to move the PCA Figure (S3) from the supplementary material into the main manuscript. The PCA analysis provides valuable insights that are central to the interpretation of the study’s results. Additionally, a more detailed discussion of the PCA results is necessary to highlight the key patterns, relationships, and trait variances observed, and to better connect these findings to the broader context of the study.
Specific Comments:
Lines 42-44: The description of drought is too broad. Specify whether the drought was terminal (during the grain filling stage), early-stage, or intermittent to give a clearer picture of the experimental setup. Mention the timing and severity of drought more explicitly to provide context for the study results.
Lines 75-76: The statement about low heritability and genotype-by-environment interactions needs further elaboration. It doesn't clarify how this challenge was addressed in the study.
Line No. 79-80: The tone is appropriately scientific, but there are a few awkward phrasings that could be revised, such as "These traits are useful as indirect selection criteria for grain yield if they show enough correlation with grain yield." This could be simplified and made more concise.
Lines 96-102: While the introduction touches on the importance of root traits for drought adaptation, the justification for focusing on roots in this study could be stronger. There's no clear linkage between previous research findings and the rationale for selecting root traits here. Provide more context on why root traits, specifically shallow and profuse systems, were chosen based on previous work or empirical evidence.
Lines 97-105: Clearly articulate the specific gaps in research on root traits or durum wheat drought tolerance that this study aims to fill.
Lines 106-110: The study's aims are general and somewhat unfocused. They could be more precise to align with the methods and outcomes. Refine the objectives to make them more specific, such as focusing on quantifying certain agronomic traits under drought conditions.
Line No. 138-139: There is also redundancy in phrases such as "was recorded as average values of 10 random plants per plot, at maturity." This could be more succinctly written as "the average height of 10 randomly selected plants per plot was recorded at maturity."
The authors refer to "leaf is shaped like a V" (line 145) when describing leaf rolling, but a more technical term like "partial leaf closure" would be more appropriate.
Line 322: The phrase "we were fortunate to have two highly contrasting seasons" could be revised to a more scientific tone, focusing on how these conditions allowed a robust evaluation of drought tolerance.
Lines 326-328: The discussion of biomass and thousand kernel weight (TKW) reductions under drought is well-placed, but it would benefit from a more detailed mechanistic explanation. For example, mention of specific physiological processes impacted by drought (e.g., reduced photosynthesis, altered translocation) would deepen the analysis.
Line 330: The statement that water stress "likely affected the entire season" is vague. Strengthening this with references to specific phenological stages and the physiological mechanisms would improve clarity.
Lines 337-342: While comparisons with other studies are useful, these lines are somewhat surface-level. Providing more in-depth discussions about why these differences may have occurred—such as variations in drought timing, intensity, or genotype specificity—would make the argument more comprehensive.
Lines 393-395: The inconsistency between root traits and yield under different water regimes could be explored in greater detail. For instance, discussing the trade-offs between water uptake and energy allocation for root growth would provide a more nuanced understanding.
Lines 400-402: The focus on modern cultivars to avoid confounding effects is valid, but a deeper discussion on whether these traits might still hold relevance across landraces would strengthen the practical implications.
Lines 344-349: The contrast between landraces and cultivars would benefit from more specific explanations of the physiological or genetic differences between the two groups that contribute to observed biomass production differences.
Line 358: The inference that landraces had more unfertile tillers is speculative, as no direct measurement was taken. Including a statement acknowledging this limitation would enhance transparency.
Lines 370-375: The suggestion that landraces could improve the vegetative growth ability of modern cultivars under drought is promising. However, this should be supported by specific recommendations on how to integrate landrace traits into breeding programs.
Acknowledge the limitation regarding the absence of direct measurement of unfertile tillers in line 358 to increase the robustness of the argument.
Add more in-text citations in areas where physiological or genetic mechanisms are discussed (lines 326-330 and 393-395) to strengthen claims.

Experimental design

Experimental design seems good. However, a few adjustments could further strengthen the experimental rigor. First, it would be beneficial to detail the specific soil properties (e.g., texture, organic matter, nutrient content) at the trial site, as these can impact root growth and drought resilience and are relevant for interpreting root trait-agronomic trait relationships. Additionally, information on the timing and intensity of drought stress across the seasons would add clarity to the environmental context. Moreover, soil moisture data during key phenological stages (e.g., tillering, flowering) would add depth to the interpretation of drought effects on performance. The study's data, collected in the 2016 and 2017 cropping seasons, may not fully capture recent shifts in climate patterns, which could influence the performance of durum wheat under current drought and environmental stress conditions. It is recommended to address how the findings from these years align with or differ from present-day climatic conditions, especially considering recent trends in temperature, rainfall distribution, and drought frequency. Discussing this aspect would enhance the relevance and applicability of the results to current and future agricultural scenarios.

Validity of the findings

The validity of the findings hinges on the completeness and clarity of the data provided, as emphasised in the Basic Reporting and Experimental Design sections.

Reviewer 3 ·

Basic reporting

Language and Clarity:
The language is clear and professional, though some areas could benefit from improved readability, particularly in the methodology and results sections. This would help readers less familiar with agronomic research follow the study quickly.

Introduction and Background:
The article provides adequate background on the importance of drought resistance in durum wheat and contextualizes the study well within the Mediterranean region. References are relevant and recent, aligning the study’s objectives with existing knowledge gaps in drought adaptation.

Structure and Figures:
The structure adheres to scientific norms, with clearly defined methods, results, and discussion sections. The figures are well-labeled and pertinent to the findings, though some could be more descriptive, particularly in representing genotype responses under drought conditions. To facilitate interpretation, figures (2 & 5) with complex statistical data could benefit from additional labeling or brief descriptions.

Experimental design

Research Scope and Questions:
The study addresses the effects of root traits on drought resistance in durum wheat. The research question is clearly stated and relevant, highlighting the need to investigate how these traits correlate with grain yield under drought.

Methodological Detail and Reproducibility:
The methods are detailed, including information on the experimental design, field setup, and statistical analysis. However, certain details could improve the understanding. For example, specifying soil type, the rationale for trait selection, and adjustments made for environmental variability (other than drought) would enhance clarity. Moreover, the study could benefit from discussing potential limitations or biases introduced by using specific genotypes that may not be fully representative of regional genetic diversity.

Validity of the findings

Data analysis
The statistical analyses are thorough, using ANOVA to explore genotypic differences. However, multiple root traits were measured, so it would benefit from multiple comparisons (MANOVA/ Repeated measure ANOVA) and correlations. This would improve confidence in the robustness of the results.

The raw data were mentioned, but more information on data availability or links to datasets could improve transparency for readers.

Conclusions
Conclusions are logically tied to the study objectives, showing that durum wheat's drought resilience can be enhanced by selecting specific root traits. However, some conclusions could be stated more cautiously, particularly around the generalizability of findings to other Mediterranean regions. Discussing how specific environmental conditions in Algeria may limit generalizability would make the conclusions more transparent.

Additional comments

Mention how the present study findings are helpful for breeding programs in more detail, especially regarding practical recommendations for breeding strategies based on root traits.

---

## Round 0.2 · accepted · Accept

Dear Authors,
I am pleased to inform you that the manuscript has improved after the last revision and can be accepted for publication.

Congratulations on accepting your manuscript, and thank you for your interest in submitting your work to PeerJ.
With Thanks

·

Basic reporting

The authors have thoroughly addressed all of my previous comments, and the manuscript is now suitable for acceptance.

Experimental design

The experimental design is appropriate

Validity of the findings

The findings are supported by robust data.

Reviewer 2 ·

Basic reporting

The authors have adequately addressed most of the queries raised in the previous round of revision, and the manuscript has been substantially improved. The article entitled "Enhancing Drought Resilience in Durum Wheat: Effect of Root Architecture and Genotypic Performance in Semi-Arid Rainfed Regions" is now suitable for acceptance.

Experimental design

Experimental design sounds promising.

Validity of the findings

Findings seems valid.

Additional comments

Grammar and clarity can be improved further during proof reading.